# Impact of Cerebral Amyloid Angiopathy in Two Transgenic Mouse Models of Cerebral β-Amyloidosis: A Neuropathological Study

**DOI:** 10.3390/ijms23094972

**Published:** 2022-04-29

**Authors:** Paula Marazuela, Berta Paez-Montserrat, Anna Bonaterra-Pastra, Montse Solé, Mar Hernández-Guillamon

**Affiliations:** Neurovascular Research Laboratory, Vall d’Hebron Research Institute, Universitat Autònoma de Barcelona, 08035 Barcelona, Spain; marazuela.paula@gmail.com (P.M.); berta.paez.montserrat@gmail.com (B.P.-M.); annabonaterra95@gmail.com (A.B.-P.); montserrat.sole@vhir.org (M.S.)

**Keywords:** APP23, 5xFAD, cerebral β-amyloidosis, preclinical MRI, cerebral microbleeds

## Abstract

The pathological accumulation of parenchymal and vascular amyloid-beta (Aβ) are the main hallmarks of Alzheimer’s disease (AD) and Cerebral Amyloid Angiopathy (CAA), respectively. Emerging evidence raises an important contribution of vascular dysfunction in AD pathology that could partially explain the failure of anti-Aβ therapies in this field. Transgenic mice models of cerebral β-amyloidosis are essential to a better understanding of the mechanisms underlying amyloid accumulation in the cerebrovasculature and its interactions with neuritic plaque deposition. Here, our main objective was to evaluate the progression of both parenchymal and vascular deposition in APP23 and 5xFAD transgenic mice in relation to age and sex. We first showed a significant age-dependent accumulation of extracellular Aβ deposits in both transgenic models, with a greater increase in APP23 females. We confirmed that CAA pathology was more prominent in the APP23 mice, demonstrating a higher progression of Aβ-positive vessels with age, but not linked to sex, and detecting a pronounced burden of cerebral microbleeds (cMBs) by magnetic resonance imaging (MRI). In contrast, 5xFAD mice did not present CAA, as shown by the negligible Aβ presence in cerebral vessels and the occurrence of occasional cMBs comparable to WT mice. In conclusion, the APP23 mouse model is an interesting tool to study the overlap between vascular and parenchymal Aβ deposition and to evaluate future disease-modifying therapy before its translation to the clinic.

## 1. Introduction

Cerebral Amyloid Angiopathy (CAA) is a major cause of lobar intracerebral hemorrhage (ICH) in the elderly, as well as an important contributor to age-related cognitive decline [1,2,3]. CAA is strongly associated with the occurrence of lobar ICH and other specific hemorrhagic magnetic resonance imaging (MRI) markers of small vessel disease, including lobar cerebral microbleeds (cMBs) and cortical superficial siderosis [4]. The most common form of CAA results from the accumulation of amyloid-beta (Aβ) protein in the walls of cerebral arteries, arterioles, and capillaries, leading to smooth muscle cell degeneration and cerebrovascular dysfunction [5,6]. The prevalence of CAA pathology increases in the presence of dementia and is particularly high in patients with Alzheimer’s disease (AD) [7,8]. Remarkably, whereas epidemiological studies have reported higher AD risk in females [9,10,11], no sex-specific differences in CAA have been described [12,13]. AD is the most common cause of dementia and it is characterized by the deposition of extracellular Aβ (neuritic plaques) and the intracellular accumulation of hyperphosphorylated tau protein (neurofibrillary tangles) [14]. Aβ peptide derives from the proteolytic cleavage of the amyloid precursor protein (APP) by β- and γ-secretases, which results in peptides typically 40 or 42 amino acids in length (Aβ40 or Aβ42, respectively) [15,16]. Aβ42 peptides are mainly deposited in neuritic plaques, whereas Aβ40 is the predominant peptide accumulated in the cerebral vasculature [17,18]. The shared role of cerebral Aβ deposition appears as a crucial event in both CAA and AD pathologies, although the overlap between these two diseases is not completely elucidated [19]. In this regard, the interactions between both pathologies have several implications for future treatments- Although anti-Aβ antibodies have been widely tested in AD clinical trials, their success has been limited [20,21,22,23], partially due to the incidence of vascular complications detected as MRI abnormalities (ARIA, Amyloid Related Imaging Abnormalities). The mechanisms that underlie ARIA are not fully understood, but available evidence suggests that the antibody-mediated breakdown of neuritic plaques releases Aβ that is then deposited in vessels, leading to increased CAA and impaired perivascular clearance [19]. Although there is debate about the clinical implication of ARIA, the use of immunotherapy in AD, and especially in CAA, has been delayed because of the occurrence of ARIA. In addition, currently, no proven effective treatment is available for CAA and the only clinical measures adopted are based on ICH prevention to avoid its recurrence [24,25].

Over the past years, efforts in the field of cerebral β-amyloidosis have been focused on the development of animal models that mimic the aspects of sporadic AD-like pathology to understand the underlying pathophysiological mechanisms and to evaluate the safety and efficacy of therapeutic drugs before the transition to the clinic. However, CAA models have been less explored. Since rodents do not spontaneously develop AD or CAA, a variety of transgenic mice models primarily overexpressing the human APP or presenilin-1 (PS1) genes has been generated harboring familial Alzheimer’s disease mutations (FAD) [26,27,28] or familial CAA mutations [29,30,31]. Our hypothesis is that the characterization of the progress of both the AD and CAA pathological hallmarks in different mouse models of cerebral β-amyloidosis will provide important insights for the study of the overlap and disparities between both diseases, which might have implications for the accuracy in the diagnosis and offer specific therapeutic opportunities for CAA or AD in the future. In particular, in this study we aimed to compare the neuropathological features of CAA in the following two widely-known transgenic mice models of cerebral β-amyloidosis: the APP23 and 5xFAD transgenic mice. APP23 transgenic mice have a 7-fold overexpression of the human APP751 isoform bearing the pathogenic Swedish mutation (K670N/M671L) under the control of the murine brain and neuron-specific thymocyte antigen-1 (Thy-1) promoter [32]. On the other hand, 5xFAD mice overexpress the human APP695 isoform and the human PS1 with a total of five AD-linked mutations, as follows: the Swedish (K670N/M671L), Florida (I716V), and London (V717I) mutations in APP, and the M146L and L286V mutations in PS1, under the control of the murine Thy-1-promoter [33]. Regarding the functional phenotype of these models, behavioral and cognitive studies in the APP23 line described the appearance of the first deficits in spatial memory at 3 months of age, which increased progressively with age [34]. However, deficits in the other types of memory did not appear until 12 months of age, although non-linear age-and genotype-dependent behavioral signatures have been found in long-lived APP23 mice [35]. On the contrary, in the 5xFAD model, important cognitive abilities were described in spatial memory tests much earlier in 3–6-month-old mice, worsening with age. In addition, 5xFAD mice from 9–12 months of age showed a progressive decrease in anxiety and motor deficits [36].

The first purpose of this study was to evaluate the progression of parenchymal Aβ load and CAA burden in the APP23 and 5xFAD mice models according to age and sex. Next, the specificity of Aβ peptides (Aβ40/Aβ42) present in plaques or cerebral vessels was studied. Finally, vascular damage was determined by analyzing the presence of cerebral microbleeds (cMBs) detected by MRI in brains from aged 5xFAD and APP23 mice. The study and comparison of the two models will allow a more adequate selection of the type, age, and sex of experimental cohorts to carry out more precise preclinical studies according to the target in each case.

## 2. Results

### 2.1. Levels of Parenchymal Aβ Deposition

Sagittal brain sections of male and female APP23, 5xFAD, and WT mice were examined in order to evaluate the differences in parenchymal amyloid deposition with age. We first confirmed the fibrillary composition of Aβ deposits by ThS staining (Figure 1A). As expected, Aβ deposition was not detected in brains from old WT mice. Furthermore, while amyloid deposits were mainly observed in the cerebral cortex, hippocampus, and thalamus regions in APP23 mice, Aβ plaques in 5xFAD mice were also detected in the midbrain, basal ganglia, and hypothalamus at advanced ages, confirming the abundant amyloid pathology in this transgenic mice model (Figure 1A). The quantification of extracellular Aβ deposits revealed a significant age-dependent increase in both APP23 and 5xFAD mice, showing the first deposits at 4 and 12 months old, respectively (Figure 1B). Remarkably, an exacerbated Aβ burden was found in both male and female 5xFAD mice, compared with APP23 mice in all the age groups studied. In fact, 4-month-old 5xFAD mice presented a similar number of Aβ deposits to 16-month-old APP23 mice, confirming, again, the massive amyloid burden in 5xFAD mice (Figure 1B).

On the other hand, the effect of sex was further evaluated in both transgenic mice models. Indeed, when the data was individually analyzed for each model, an increased number of amyloid ThS positive deposits were found in female APP23 mice compared to males during aging (Figure 2A). In contrast, sex differences in the number of Aβ deposits were only observed in young 5xFAD mice at 4 and 8 months old (Figure 2A). Interestingly, Aβ deposits in 5xFAD were smaller but more numerous than those observed in APP23 mice (Figure 2A,B). However, while the size of amyloid deposits was very similar in 5xFAD mice at all ages studied, a significant increase in size with aging was observed in APP23 mice, with a greater increase in the older females compared to age-matched males (Figure 2B). Indeed, although the 5xFAD brains presented a higher number of deposits, the area occupied by those deposits was much smaller than in APP23 mice (Figure 2C). Accordingly, the total area covered by ThS deposits was strongly dependent on age but also modulated by sex in APP23 mice, whereas in 5xFAD mice the main differences were attributed to the age factor (Figure 2C). Altogether, these results suggest a more prominent sex dependent parenchymal Aβ pathology in APP23 than in 5xFAD mice.

### 2.2. Levels of Vascular Aβ Deposition

To assess cerebrovascular amyloid deposition in the brain of APP23, 5xFAD, and WT mice, resorufin staining was performed. As shown in Figure 3A, no Aβ-positive vessels were detected in WT or 5xFAD mice at any age. As expected, leptomeningeal and cortical vessels from APP23 mice were strongly positive for resorufin staining, which specifically binds to cerebrovascular Aβ (Figure 3A). The quantification of CAA burden in APP23 mice revealed an age-related increase in vascular Aβ deposits, detecting the first Aβ-positive vessels at 12 months old (Figure 3B). Interestingly, in contrast to parenchymal Aβ deposition, no differences in CAA-affected vessels were found between APP23 males and females, although a trend towards a higher number of Aβ positive vessels was observed in 24-month-old males compared to age-matched females (Figure 3B).

### 2.3. Analysis of Aβ40 and Aβ42 Deposition

We further determined the specificity of the Aβ peptide present in plaques or cerebral vessels from 16-month-old 5xFAD and APP23 mice. As shown in Figure 4, the vast majority of parenchymal deposits consisted in compact amyloid, with dense-cored plaques throughout the cortex in both transgenic models. Diffuse amyloid deposits were also observed at this age, although they were less abundant. The presence of total Aβ- positive leptomeningeal and cortical vessels was then confirmed by 4G8 immunostaining in APP23 mice (Figure 4A). Remarkably, although leptomeningeal and parenchymal Aβ deposits were positive for both Aβ42 and Aβ40 antibodies, the immunodetection of Aβ40 was more pronounced (Figure 4(A5–A7)), confirming the predominance of Aβ40 pathology over Aβ42 (Figure 4(A8–A10)) in APP23 mice. Furthermore, Aβ cortical vessels were stained with Aβ40-specific antibody in APP23 mice (Figure 4(A6)), whereas no immunodetection for Aβ was apparent in cortical vessels or capillaries from 5xFAD mice (Figure 4(B3–B9)). However, occasional leptomeningeal vessels from aged 5xFAD mice showed some patchy and scarce Aβ40 and Aβ42 immunodetection (Figure 4(B5,B8)), although this staining was much less prominent than in APP23 mice (Figure 4(A5,A8)). Interestingly, some positive staining was observed in capillaries using the 4G8 antibody in APP23 mice (Figure 4(A1)), confirming that it can be used as a suitable model for the study of CAA type 1 [37].

### 2.4. Determination of CAA-Related Cerebral Microbleeds

To further evaluate the neuropathological contribution of CAA, ex vivo MRI was performed in 20-month-old male APP23 and 5xFAD mice and WT littermates to specifically detect cerebral microbleeds. Representative T2* images of cMBs in APP23, 5xFAD, and WT mice are shown in Figure 5A. Remarkably, APP23 mice displayed a substantially higher number of cMBs compared with age-matched 5xFAD mice and WT littermates (Table 1 and Figure 5A). In this regard, the distribution of micro hemorrhages in APP23 mice was mainly cortical, with only two cMBs in deep regions among all the cMBs identified in this transgenic cohort. On the other hand, although aged 5xFAD and WT mice also presented occasional cortical cMBs, the total number did not significantly differ between them (Table 1). In particular, only one WT mice showed a single cortical cMB and 50% of the 5xFAD individuals included in the study presented one or two cMBs (Figure 5B). In contrast, 42.8% of APP23 mice showed from three to five cMBs, whereas the other 57.2% of APP23 mice presented more than five cMBs (Figure 5B). Although the size and volume of the lesions were very similar between the groups, the total hemorrhagic area was significantly different among them due to the greater number of total cMBs in APP23 mice (Table 1). Additionally, Prussian blue staining evidenced the presence of blood extravasation in APP23 mice, confirming the black or hypointense lesions detected by T2*-MRI (Figure 5C).

## 3. Discussion

The discovery of AD and CAA familial mutations in genes involved in Aβ processing and the identification of Aβ in amyloid plaques in AD patient’s brains has led to the well-known hypothesis that Aβ is the principal pathological feature of AD and CAA [29,38,39,40]. The amyloid hypothesis proposes that these diseases are caused by an imbalance between Aβ production and clearance leading to parenchymal and vascular Aβ deposits [41], although the precise mechanisms explaining Aβ brain accumulation remain elusive. Several pathways have been described to remove Aβ from the brain, including active transport across the BBB and perivascular and lymphatic drainage [42]. In this regard, the impairment of the intramural periarterial drainage (IPAD) pathway seems to play a fundamental role in CAA [43,44].

Over the last decades, the development of AD therapies has been focused on targeting the Aβ peptide and promoting its clearance. Nevertheless, the high failure rate of these therapies in clinical trials, and in particular, of anti-Aβ drugs, highlights the urgent need to develop novel disease-modifying treatments in the field of cerebral β-amyloidosis. In this context, although Aβ can be detected in vivo by positron emission tomography (PET) amyloid tracers, no approach so far is specific enough to provide a differential diagnosis between CAA and AD [45], which also makes the selection of subjects difficult in clinical trials. Thus, animal models are an essential tool for understanding the neuropathological mechanisms underlying AD and CAA. Elucidating the impact of genetic risk factors such as age or sex, and the contribution of vascular dysfunction in these experimental models, could lead to the development of new specific therapies for AD and/or CAA, improve the accuracy of the current diagnosis, and even facilitate the discrimination between vascular and parenchymal Aβ deposition in AD and CAA pathologies.

In our study, we evaluated some of the neuropathological consequences of Aβ deposition in the APP23 and 5xFAD transgenic mice models in terms of age and sex. APP23 mice overexpress the human APP751 isoform harboring a single FAD mutation, the Swedish double-mutation (K670N/M671L) [32]. This model was selected for its previous contribution to the study of cerebrovascular pathology associated to Aβ deposition [28,46,47]. In contrast, 5xFAD mice develop early-onset parenchymal amyloid plaques and it was selected to be considered a common model for the study of AD [36,48]. 5xFAD mice overexpress the human APP691 isoform with three FAD mutations (K670N/M671L, I716V, and V717I) and the human PS1 with two FAD mutations (M146L and L286V) [33]. Although this model recapitulates the main features of amyloid AD-like pathology, no causes of human AD are explained by multiple FAD mutations. Therefore, although they have provided valuable insight into AD pathogenesis, an accurate interpretation of the results should be performed before the translation to human studies.

We first confirmed the progressive increase in parenchymal amyloid deposits in both 5xFAD and APP23 mice with age, detecting the first neuritic plaques at 4 and 8 months old, respectively. Consistent with previous studies, an exacerbated Aβ pathology was found in 5xFAD mice when compared to APP23 mice, although amyloid plaques were smaller in 5xFAD mice. In this regard, sex-specific differences were observed in APP23 mice in all the age-points studied, with a higher plaque pathology load in females compared to males. This result is aligned with a recent study in which stronger amyloidosis and astrogliosis were reported in female APP23 mice [49]. Notably, whereas 5xFAD mice at advanced ages did not show sex-related differences, 4- and 8-month-old females presented increased Aβ deposition compared to age-matched males. Indeed, the 5xFAD original publication, in which mice were created on a B6SJL hybrid background, reported a trend towards greater plaque deposition in females [33], and interestingly, Bundy et al. [50] have recently described that molecular AD pathology was more profound in the hippocampus transcriptome of 4-month-old female 5xFAD mice compared to males. In humans, it is widely known that along with age and the APOE4 genotype, female sex is a major risk factor for developing human AD [51]. In addition, neuropathological studies in AD post-mortem brain tissues reported more severe AD pathology in females, with more extensive neuritic plaque deposition throughout the brain compared to men [52,53], confirming previous results found in pre-clinical mouse models. It is important to note that although sex steroids and hormones have been extensively proposed as key contributors to sex-related differences in AD [54,55,56], the potential mechanisms influencing this sex bias are not fully understood and are currently under investigation [56]. It has been proposed that after menopause, the decline in estrogens was linked to AD progression [57]. However, a large clinical trial in women found that estrogen-based hormone therapy was associated with increased risk rather than protection from dementia [58]. Alternatively, clinical observations in men showed that the age-related loss of androgens was associated with circulating and brain Aβ levels, suggesting that reduced androgens may contribute to AD pathology [59]. Indeed, in vitro and in vivo studies have also demonstrated that testosterone reduces Aβ deposition in AD models [60,61]. Taken together, future research should focus on exploring the mechanism underlying sex differences to develop effective strategies for the diagnosis and treatment of AD. Our results using experimental models highlight the translational use of transgenic mice, providing important evidence on sex-specific differences in AD progression as occurs in humans.

Due to the high overlap between CAA and AD and the shared role of Aβ in both pathologies, we further explored the amyloid deposition along the cerebral vasculature of these two transgenic mouse models. Consistent with previous studies, we confirmed the prominent vascular amyloid deposition in the APP23 model with age [32,37,46]. Moreover, the immunostaining with Aβ40 and Aβ42 specific antibodies in our study corroborated the predominance of Aβ40 staining. Therefore, the increased Aβ40:Aβ42 ratio in APP23 mice may lead to vascular amyloid deposition. In this regard, the Aβ42 peptide is less soluble and more likely to be retained in the parenchyma, whereas Aβ40 is more soluble and can diffuse along perivascular pathways to accumulate in the walls of vessels [19]. It should be pointed out that while female APP23 mice displayed higher parenchymal Aβ deposition, no apparent differences were found in the CAA load between female and male mice. This result is in line with human CAA studies, in which sex differences have not been reported and sex-specific bias remains unclear [12,13]. In this regard, we hypothesized that parenchymal Aβ clearance could be modulated by sex-specific mechanisms, whereas the clearance of vascular amyloid may not. However, as mentioned above, further studies are mandatory to understand the contribution of sex in Aβ-related pathologies.

On the other hand, no Aβ-positive vessels were detected in the vasculature from 16-month-old 5xFAD mice with the specific resorufin staining. Notably, although cortical vessels did not show any immunoreactivity for Aβ40, some leptomeningeal arteries from aged mice were stained patchily with both Aβ42 and Aβ40 antibodies. In this regard, the study of the contribution of CAA pathology in 5xFAD mice has been scarce and controversial. To the best of our knowledge, there is only one study reporting CAA in leptomeningeal and penetrating vessels from 3- to 5-month-old 5xFAD mice by in vivo two- photon microscopy [62], but the volume of CAA reported was highly variable between mice, observing no CAA in one of the five mice studied. However, another study previously reported the absence of CAA in 5xFAD mice up to 9 months old using classical immunostaining techniques [63]. This same study found higher soluble and insoluble Aβ42 cerebral levels than Aβ40 in this transgenic mouse [63]. Despite the mutations carried by 5xFAD mice favoring the production of Aβ42, our immunohistochemistry results showed a moderate Aβ42 and Aβ40 positivity in aged 5xFAD brains. A possible explanation could be the accumulation of truncated Aβ species and pyroglutamate-modified Aβ peptides in the brains of these mice, as was previously reported by Jawhar et al. [36]. Nevertheless, our findings should be further explored using complementary biological methodology. In conclusion, although in both APP23 and 5xFAD mice the source of non-mutated human Aβ peptides produced is through neuronal expression [32,33,37], the generation and velocity of the peptides, defined by the mutations in the APP and/or PS1 genes, seem to determine the length and post-translational modifications of the peptides, which would influence the final localization of the Aβ deposition within the brain.

Finally, to assess the clinic phenotype of CAA, we evaluated the occurrence of cMBs in aged transgenic mice by MRI. In this context, it has been proposed that the accumulation of Aβ in human cerebral vessels leads to smooth muscle degeneration, blood-brain barrier (BBB) dysfunction, and eventually to vessel rupture and blood leakage, visualizing cMBs by MRI [3,6]. The cMBs in the lobar location represent one of the main neuroimaging findings in CAA patients and are associated with an increased risk of ICH recurrence [64]. The present study confirmed previous findings in which cMBs were detected in APP23 mice [65,66]. Indeed, an age-dependent increase in the number of cMBs in APP23 mice in vivo from 16 months old was previously reported, showing a size distribution comparable to our ex vivo data (diameter 200–300 μm) [45,65,66]. Although microvascular alterations can be also detected by magnetic resonance angiography in this transgenic model [67], further studies are needed to understand the mechanisms explaining BBB dysfunction and vascular leakage in APP23 mice. Altogether, we can conclude that the APP23 model constitutes an invaluable model to study the clinical consequences of CAA pathology compared to familial CAA transgenic models in which the first hemorrhages appear at very late-onset (APPDutch mice) [68] or are very rare (SwDI mice) [69]. Despite the exacerbated amyloid pathology observed in 5xFAD mice, we were only able to detect a few sporadic cMBs in 5xFAD brains. Our results did not show differences regarding the number of cMBs between 5xFAD and WT mice. Presumably, it may be due to the absence of marked CAA pathology in these mice, as we have neuropathologically demonstrated in the present study, although the relatively low number of animals used for the MRI analysis is a limitation to consider. Remarkably, 5xFAD mice harboring the human APOE alleles (E3 and/or E4) exhibited cMBs related to CAA pathology [70,71], potentially due to the widely-known impact of the APOE gene as a risk factor for human CAA and CAA-related hemorrhage [72,73,74]. In that study, the authors suggested that cMBs precede Aβ plaques and might seed their formation. In contrast with their hypothesis, our results demonstrate that cMBs constitute a late event in cerebral β-amyloidosis in APP23, suggesting that they arise from the CAA-damaged vasculature.

In summary, an age-dependent vascular and parenchymal Aβ deposition was confirmed in APP23 mice, with a greater increase in Aβ-plaques presence in females. We corroborated the clinical phenotype of CAA with the presence of multiple hemorrhages by MRI. In contrast, in 5xFAD mice, CAA pathology was almost absent in this model, whereas an exacerbated parenchymal Aβ load was observed. Thus, beyond gaining insights into the mechanisms favoring parenchymal or vascular Aβ accumulation, the APP23 transgenic model may also be useful to improve the understanding of the vascular contribution in AD pathogenesis and to pre-clinically evaluate the potential therapeutic strategies in CAA. In conclusion, we consider that the characterization of the commonly used experimental models in the AD and CAA field is indispensable to ensure a greater translation of pre-clinical results to human clinical trials. The availability of a wide variety of mice models with different phenotypes raises a good opportunity to confirm or discard pre-established hypotheses in the field of cerebral β-amyloidosis.

## 4. Material and Methods

### 4.1. Transgenic Animals

All animal procedures were approved by the Ethics Committee for Animal Experimentation of the Vall d’Hebron Research Institute and conducted in compliance with Spanish legislation and with the Directives of the European Union. The APP23 mice (B6.Cg-Tg (Thy1-APP) 3Somm/J) were purchased from The Jackson Laboratory (Bar Harbor, ME, USA) and the 5xFAD mice (B6.Cg-Tg (APPSwFlLon, PSEN1*M146L*L286V) 6799 Vas/ Mmjax) were generously provided by Dr. Santiago Rivera, Marseille University, France. The male hemizygous APP23 and 5xFAD mice with the same genetic background (C57BL/6) were backcrossed with the female C57BL/6 mice (Janvier Labs, Le Genest-Saint-Isle, France), and the genotype was tested by Transnetyx (Cordova, TN, USA). The male and female APP23 and 5xFAD mice and the wild-type (WT) littermates were aged in the animal facility of our institution to obtain the final study cohort. All mice selected for the study were housed with same-sex individuals from their weaning and not used for breeding purposes. Mice were housed in a climate-controlled environment on a 12/12 h light/dark cycle with food and water ad libitum. The number of mice per group is specified in each figure.

### 4.2. Tissue Processing for Histological Analysis

For tissue collection, mice were deeply anesthetized under isoflurane flow and transcardially perfused with 25 mL of ice-cold phosphate-buffered saline (PBS). The brains were rapidly removed from the skull and divided into hemispheres. One hemisphere was frozen in liquid nitrogen and stored at −80 °C until further processing for biochemical analyses, and the other hemisphere was fixed in 10% formalin for 48 h before paraffin embedding.

### 4.3. Tissue Processing for Magnetic Resonance Imaging

The transgenic male APP23 and 5xFAD mice and WT littermate controls were aged to 20 months and euthanatized to perform ex vivo brain magnetic resonance imaging (MRI). Briefly, animals were deeply anesthetized and transcardially perfused with 40 mL of ice-cold PBS followed by 40 mL of 4% paraformaldehyde (PFA). The brains were carefully removed from the skull, immersed in PFA for 72 h, and then rehydrated in PBS preserved with 0.01% sodium azide at 4 °C for a minimum of 10 days for tissue stabilization.

### 4.4. MRI Protocol and Analysis

All post-mortem brains were scanned on a 7T horizontal magnetic system (Bruker BioSpec 70/30 USR, Ettlingen, Germany) in the joint nuclear magnetic resonance facility of the Universitat Autònoma de Barcelona. Samples were immersed in Galden^®^D05 (Solvay, Bollate, Italy), a fluorinated liquid, to reduce susceptibility artefacts and placed into a custom-built brain holder for imaging. T2*-weighted images were obtained in coronal planes with the operational software Paravision (Bruker) using the following parameters: repetition time 700 ms, effective echo time 8 ms, matrix size 160 × 160, field of view 1.28 × 1.28 cm^2^, slice thickness 0.3 mm, and number of slices 28. The total acquisition time for a voxel of a size of 0.08 × 0.08 × 0.3 mm3 was 22 min. For MRI quantification, hypointense signals on T2* were counted as hemorrhages and classified as cerebral microbleeds (50–300 μm diameter) or larger hemorrhages (>300 μm). To avoid counting the same hemorrhagic lesion multiple times, its presence was carefully controlled over several consecutive slices. After ex vivo scanning, all brains were fixed and paraffin-embedded for future tissue histological examinations.

### 4.5. Thioflavin S Staining

The fibrillar Aβ deposits were quantified by Thioflavin S (ThS) staining in sagittal brain paraffin sections of APP23 and 5xFAD mice and littermate controls for both sexes at different ages (from 4 to 24 months old). After deparaffinization for 1 h at 65 ºC and rehydration, sections were immersed in 1% ThS solution (Sigma-Aldrich, Saint Louis, MO, USA) dissolved in 75% ethanol for 30 s. The excess of ThS was removed and slices were re-immersed in 0.1% ThS for 1 min, dehydrated, and mounted in DAPI-containing mounting media (Vector Laboratories, Burlingame, CA, USA) for nuclei counterstaining. The Pannoramic 250 scanner (3DHistech, Budapest, Hungary) was used at 20x objective to digitize the slides and images were captured using Case Viewer Software (3DHistech). To quantify the number of ThS-positive deposits, images were converted to 32-bit grayscale and the brain region of interest was delimited (whole brain section except for the olfactory bulb and cerebellum). The threshold was then adjusted to reduce the background. A common pixel threshold value was applied to all the images. Finally, the number of total ThS-positive deposits was quantified and divided by the selected area (pixels^2^). The average size of the deposits (μm^2^) and the area occupied by the deposits (%) were also calculated using NIH ImageJ software.

### 4.6. Resorufin Staining

For the specific detection of Aβ-positive vessels, resorufin staining was performed in paraffin-embedded mice brain sections following the protocol described by Han et al. [75]. Briefly, all samples were deparaffinized, washed 3 times in PBS, and permeabilized in PBS-0.2% Triton X-100 (PBST) for 30 min. Samples were then immersed in 1 mM resorufin (Sigma-Aldrich) dissolved in PBST for 5 min. After rinsing with PBS, samples were rinsed with PBS-50% ethanol for 3 min and dehydrated. Finally, DAPI- containing mounting media (Vector Laboratories) was used as contrast staining. Samples were digitized at 20× objective using the Pannoramic 250 scanner (3DHistech) and the number of positive vascular deposits were manually determined in the selected area. Data are expressed as the number of Aβ-positive deposits per area (pixels^2^). Images were captured using Case Viewer Software (3DHistech).

### 4.7. Aβ Peptide Immunohistochemistry

Immunohistochemistry was performed on consecutive 5 μm thick sagittal paraffin sections from 16-month-old female APP23 and 5xFAD mice. All samples were deparaffinized, rehydrated, and incubated with citrate buffer to improve the antigen exposition (10 mM sodium citrate, 0.05% Tween20, pH = 6) for 30 min at 95 ºC. Sections were then blocked in 10% goat serum in tris buffered saline (TBS) supplemented with 0.2% TritonX-100 (TBST, Sigma-Aldrich) at room temperature for 1 h, followed by overnight incubation with the following primary antibodies: mouse monoclonal anti-4G8 (1:5000, Biolegend, San Diego, CA, USA), rabbit polyclonal anti-Aβ40 (1:5000, #AB5074P, Millipore, Temecula, CA, USA), and rabbit polyclonal anti-Aβ42 (1:5000, #AB5078P, Millipore). After rinsing, sections were treated with 3% hydrogen peroxide for 15 min to block the endogenous peroxidases. Samples were then incubated with biotinylated anti-mouse IgG or anti-rabbit IgG (1:1000, Vector Laboratories) at room temperature for 1 h, followed by streptavidin-horseradish peroxidase (HRP; 1:1000, Vector Laboratories) incubation. Finally, diaminobenzidine (DAB; Dako, Denmark) was applied to the samples and the sections were immersed in Harris hematoxylin solution (Sigma-Aldrich) used as contrast staining. Samples were dehydrated and DPX (Sigma-Aldrich) was used as a mounting medium. Specific immunodetection was confirmed by the negative signal obtained with the incubation of the corresponding secondary antibody only (no primary antibody control). The Pannoramic 250 scanner (3DHistech) was used at 20× objective to digitize the histological slides and the images were captured using Case Viewer Software (3DHistech).

### 4.8. Prussian Blue Staining

After ex vivo imaging, brains were paraffin embedded and sliced in 5 μm thick coronal sections. Prussian blue staining was performed to detect the presence of cerebral microbleeds due to the specific staining for ferric iron and hemosiderin complexes. After deparaffinization and rehydration, the sections were stained with a commercial Prussian blue iron stain kit (Polysciences Inc, Warrington, PA, USA) following the manufacturer’s instructions. Briefly, sections were incubated with equal parts of potassium ferrocyanide, as follows: hydrochloric acid solution for 40 min, rinsed in water, and counterstained with Nuclear Fast Red for 5 min. Finally, sections were dehydrated and DPX was used as a mounting medium. The Pannoramic 250 scanner (3DHistech) was used at 20× objective to digitize the histological slides and the images were captured using Case Viewer Software (3DHistech).

### 4.9. Statistical Analysis

The SPSS 20.0 package (IBM Corporation, Armonk, NY, USA) and GraphPad Prism 6 (GraphPad Software, La Jolla, CA, USA) were used for statistical analyses. The normality was assessed by the Shapiro–Wilk test. In normally distributed variables, the significant differences between groups were assessed using Student t-test and one-way analysis of variance (ANOVA) with Bonferroni’s post hoc test when appropriate. The effects of sex (S) and age (A) were evaluated by two-way ANOVA with Bonferroni’s post hoc test for multiple comparisons. In non-normally distributed variables, the Mann–Whitney U test was assessed for one-to-one comparisons and the Kruskal–Wallis test for multiple comparisons. Data are expressed as the mean ± standard error of the mean (SEM) for normal distributions or as the median (interquartile range) for non-normal distributions. A *p*-value < 0.05 was considered statistically significant and a *p*-value < 0.1 was considered a statistical tendency.

## Figures and Tables

**Figure 1 ijms-23-04972-f001:**
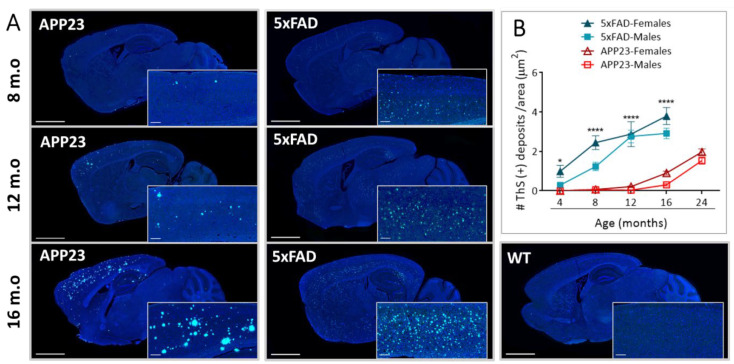
Extracellular Aβ deposition in 5xFAD and APP23 mice. (**A**) Representative images of sagittal brain sections from female 5xFAD, APP23, and WT mice showing Aβ-positive fibrillar deposits stained with Thioflavin-S (ThS). Scale bar indicates 2 mm and 200 μm; m.o: months old. (**B**) Representation of the number of parenchymal Aβ deposits in male and female 5xFAD mice and APP23 mice at different ages. Data are expressed as the number of ThS-positive deposits (fibrillar Aβ deposits) per area. *n* = 4–7/group. Statistical differences were analyzed between APP23 and 5xFAD mice (females + males) and represented as: * *p* < 0.05, **** *p* < 0.0001.

**Figure 2 ijms-23-04972-f002:**
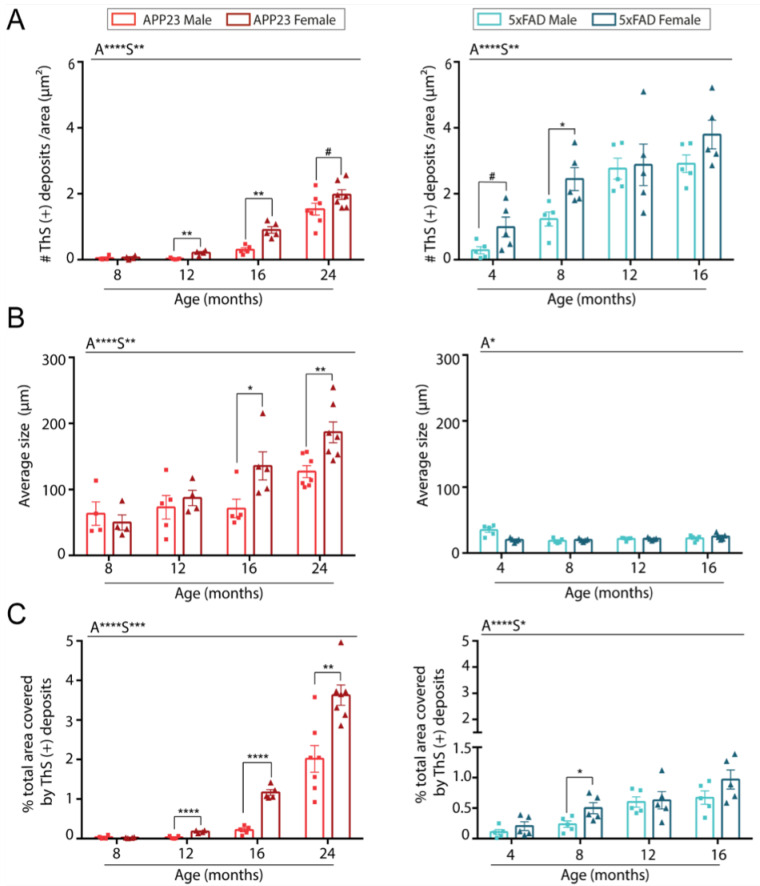
Quantification of parenchymal Aβ burden in brains from male and female 5xFAD and APP23 mice at different ages. (**A**) Quantification of the number of ThS-positive deposits corrected by area. (**B**) Quantification of the average size of ThS-positive deposits. (**C**) Quantification of total area occupied by the deposits (%). *n* = 4–7/group. The effect of sex (S) and age (A) was evaluated in each subgroup of data. Statistical differences were analyzed among groups as indicated and represented as: # *p* < 0.1, * *p* < 0.05, ** *p* < 0.01, *** *p* < 0.001, **** *p* < 0.0001.

**Figure 3 ijms-23-04972-f003:**
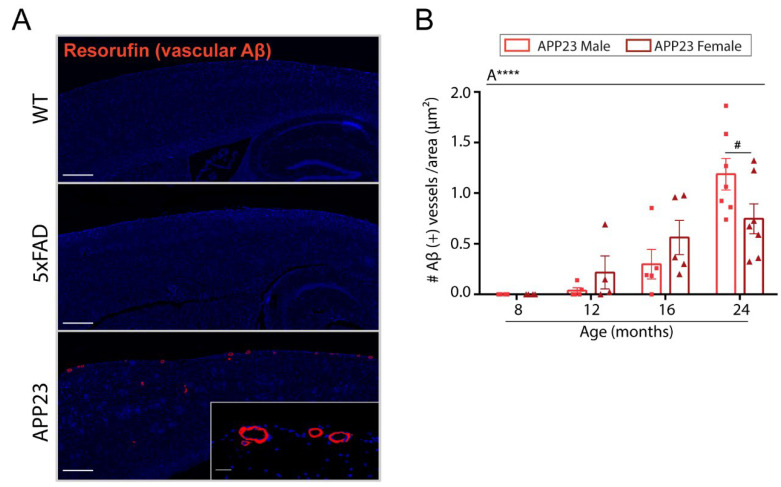
Vascular Aβ deposition in 5xFAD and APP23 mice. (**A**) Representative images of resorufin staining in 16-month-old WT, 5xFAD, and APP23 female mice. Scale bar indicates 200 μm and 20 μm, respectively. (**B**) Quantification of the number of Aβ-positive vessels identified by resorufin staining in APP23 mice. Data are expressed as the number of Aβ-positive vessels per area (pixels^2^). *n* = 4–7/group. The effect of sex (S) and age (A) was evaluated in each subgroup of data. Statistical differences were analyzed among groups as indicated and represented as: # *p* < 0.1, **** *p* < 0.0001.

**Figure 4 ijms-23-04972-f004:**
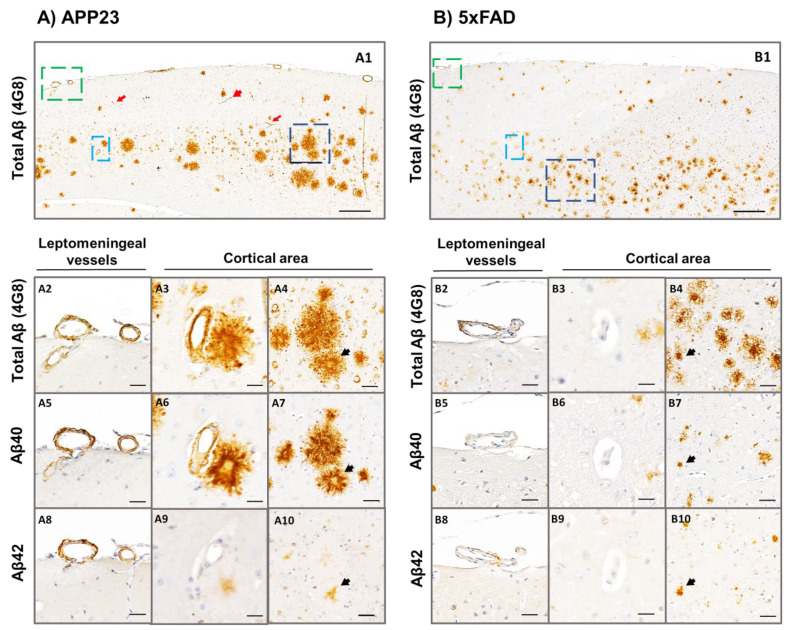
Specific Aβ40 and Aβ42 immunodetection in the parenchyma and cerebral vasculature in APP23 and 5xFAD mice. Total Aβ (determined by anti-4G8), Aβ40, and Aβ42 immunodetection analyzed in brain sections of: (**A**) 16-month-old female APP23:and, (**B**) 5xFAD (mice. Zoom images from leptomeningeal vessels (green), cortical vessels (cyan), and amyloid plaques (dark blue) are shown in A2–A10 panels for the APP23 model and in B2-B10 panels for the 5xFAD model. A2–A4 and B2–B4 represent consecutive brain sections stained with anti-4G8 primary antibody; A5–A7 and B5–B7 represent consecutive brain sections stained with anti-Aβ40 primary antibody; and A8–A10 and B8–B10 represent consecutive brain sections stained with anti-Aβ42 primary antibody. Red arrows in A1 indicate amyloid-affected capillaries in APP23 brains. Black arrows in A4–A10 and B4–B10 indicate the same amyloid-plaques analyzed in consecutive sections using different primary antibodies. Scale bar in A1/B1 indicates 200 μm and in A2–A10 and B2–B10 indicates 40 μm.

**Figure 5 ijms-23-04972-f005:**
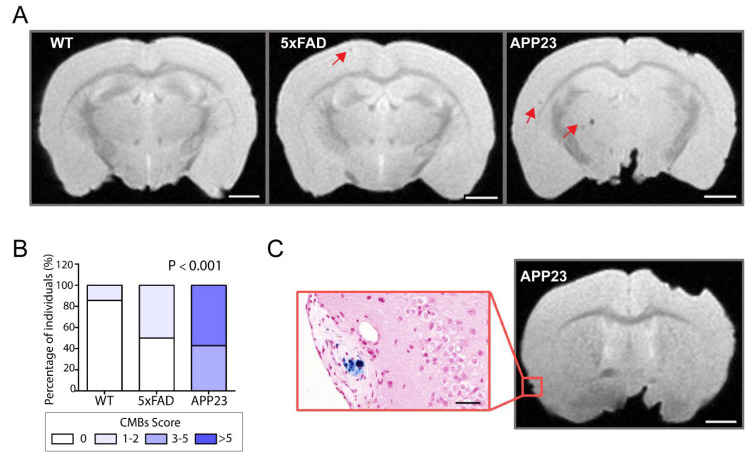
MRI detection of CAA-related cerebral microbleeds in 5xFAD and APP23 mice. (**A**) Representative T2* magnetic resonance images (MRI) of cerebral microbleeds (cMBs) in 20-month-old male WT, 5xFAD, and APP23 mice. cMBs are indicated with red arrows. Scale bar indicates 2 mm. (**B**) Distribution of cMBs in WT, 5xFAD, and APP23 mice, according to the percentage of individuals affected. N = 4–7/group. (**C**) Comparison of cMBs in APP23 mice T2* sequences and Prussian blue staining showing iron hemosiderin deposits. Scale bar indicates 2 mm and 40 μm.

**Table 1 ijms-23-04972-t001:** Cerebral microbleeds quantification in APP23, 5xFAD, and WT mice.

	WT (*n* = 7)	5xFAD (*n* = 4)	APP23 (*n* = 7)	*p*-Value
**Total cMBs**	0	0.50 (0–1.75) ^$^	6 (4–8)^**^	**<0.001**
**Size (µm)**	0	195.91 (48.44–249.25) ^**^	231.09 (200.13–261.58) ^***^	**<0.001**
**Volume (mm^3^)**	0	0.01 (0.002–0.015)	0.01 (0.009–0.016) ^***^	**<0.001**
**Hemorrhagic area (mm^2^)**	0	1.47 (0–7.22) ^$^	28.31 (14.68–79.82) ^**^	**<0.01**

Data are expressed as the median (interquartile range). cMBs, cerebral microbleeds. *p*-Values < 0.05 are shown in bold. ^**^ *p* < 0.01 vs. WT; ^***^
*p* < 0.001 vs. WT; ^$^
*p* < 0.05 vs. APP23.

## Data Availability

The data that support this study are available on request from the corresponding autor, M.H.-G.

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
