# Peer review of "Impact of Cerebral Amyloid Angiopathy in Two Transgenic Mouse Models of Cerebral β-Amyloidosis: A Neuropathological Study"

_ijms, 2022, doi:10.3390/ijms23094972_

Round 1

Reviewer 1 Report

Interesting Story but need more control data. 

I have several questions. 

  1. Ab plaque size is larger in APP23 than 5XFAD. What is the pathological meaning of this?
  2. Ab plaque accumulated in specific region in APP23 mice brain, but 5XFAD is evenly detected. what's the meaning?
  3. Do you have 2’ ab only control for all IHC and IF? Especially in Figure Vessel elastin easily stained by other Ab, so you must check 2’ ab control.
  4. Which mouse model has a human phenotype like that of humans? What human diseases cause vascular Ab accumulation?
  5. What is the mechanism by which Ab accumulates in blood vessels? If possible, please add data or add it to the discussion section.

Author Response

Interesting Story but need more control data. 

We appreciate the reviewer’s comment and we have tried to improve the manuscript by adding some necessary controls in our experiments.

1. Ab plaque size is larger in APP23 than 5XFAD. What is the pathological meaning of this?

We are not certain about the meaning of this plaque size difference between models, although we believe that it might be an outcome indirectly related to amyloid nucleation and aggregation processes. Previous reports in the literature have found that plaques do not substantially increase in size over the clinical course of the disease in human AD brain (Serrano-Pozo et al; J Neuropathol Exp Neurol. 2012). In our study, we found that while the size of amyloid deposits was very similar in 5xFAD mice at all ages studied, a significant increase in size with aging was observed in APP23 mice, with a greater increase in the older females compared to age-matched males. We consider that reporting the plaque size variable in experimental models of the disease might be important, for instance, to pre-clinically test the efficacy of anti-amyloid compounds based on the inhibition of aggregation and protein misfolding. Thus, establishing the proper model in terms of plaque size, as well as selecting the appropriate sex and age, might be relevant factors to design successful future studies.

2. Ab plaque accumulated in specific region in APP23 mice brain, but 5XFAD is evenly detected. What’s the meaning?

We consider that the specific areas where Aβ accumulates in the APP23 mice resemble the CAA/AD human pathology. However, in the case of 5xFAD mice, there is a fast overproduction of amyloid due to the five AD-linked mutations in this model. Although this mice line recapitulates some features of amyloid AD-like pathology, no causes of human AD are explained by the presence of all of these mutations. Therefore, although the 5xFAD model has provided valuable insight into AD pathogenesis, an accurate interpretation of the results, especially considering the evolution and distribution of Aβ deposits, should be done before the translation to human studies.

3. Do you have 2’ ab only control for all IHC and IF? Especially in Figure Vessel elastin easily stained by other Ab, so you must check 2’ ab control.

When an IHC using a new antibody is set up in the lab, a control with the incubation of only secondary antibody is always performed. In the present study, ICH was performed using the following primary antibodies: mouse monoclonal anti-4G8 (1:5000, Biolegend, San Diego, CA, USA), rabbit polyclonal anti-Aβ40 (1:5000, #AB5074P, Millipore, Temecula, CA, USA) and rabbit polyclonal anti-Aβ42 (1:5000, #AB5078P, Millipore). Thus, control analysis performed through the incubation of brain sections with biotinylated anti-mouse IgG or anti-rabbit IgG (1:1000, Vector Laboratories) were performed.

The incubation of secondary antibodies on brain slides from 20 months old APP23 mice, without the primary antibody step, did not induce a positive signal after DAB incubation. These methodological controls indicated the specificity of the primary antibodies (anti-4G8, anti- Aβ40 and anti- Aβ42) for parenchymal and vascular amyloid deposits showed in Figure 5 (new Fig. 4).

We have included a sentence in the Methods section of the new version of the manuscript to expose this point: “Specific immunodetection was confirmed by the negative signal obtained with the incubation of the corresponding secondary antibody only (no primary antibody control).”

On the other hand, we would like to clarify that immunofluorescence (IF) has not been conducted in the present study. Thioflavin S and Resorufin staining methodologies do not involve the incubation with secondary antibodies. We consider that the best control in these cases is the demonstration of the negative staining in wild-type brains (without an overload of Aβ), as shown in Fig. 1 and 4 (new Fig. 3).

4. Which mouse model has a human phenotype like that of humans? What human diseases cause vascular Ab accumulation?

The human disease caused by the vascular Aβ deposition in the brain is CAA (Cerebral Amyloid Angiopathy). In our study, we have clearly shown that only the APP23 transgenic model can be considered an experimental model of this pathology, as Aβ accumulation can be found in cortical and leptomeningeal arteries inducing the occurrence of cerebral microbleeds.

On the other hand, both transgenic animal models studied (APP23 and 5xFAD), recapitulate some specific pathological features of human AD pathology, such as the presence of Aβ-plaques, but not all of them. Hence, again, interpretation of results using both models has to be taken with caution when translated to humans.

5. What is the mechanism by which Ab accumulates in blood vessels? If possible, please add data or add it to the discussion section.

The amyloid hypothesis proposes that cerebral Aβ deposition is caused by an imbalance between its production and clearance leading to parenchymal and vascular Aβ accumulation, although the precise mechanisms explaining Aβ brain accumulation remain elusive. Several pathways have been described to remove Aβ from the brain, including active transport across the BBB and perivascular and lymphatic drainage (Tarasoff-Conway et al., Nat Rev Neurol. 2015). In this regard, the impairment of the intramural periarterial drainage (IPAD) pathway seems to play a fundamental role in the vascular Aβ accumulation leading to CAA development (Weller et al. Brain Pathol. 2008; Hawkes et al. Brain Pathol. 2014).

As requested by the reviewer, a paragraph referred to this point has been included in the Discussion section of the new version of the manuscript.

Reviewer 2 Report

Authors compared the pathological characters of APP23 mouse model and 5xFAD mouse models. According to the results, authors concluded that APP23 is an interesting model since the APP23 mouse models have CAA and plaques while 5xFAD only showed plaques accumulation in brian. 

Firstly, both mice models have been used in AD studies for years, the characters of both models have already been recorded well and bred with other transgenic mouse models for new characters. Tons of papers have been published. Besides the two models, there are also a lot of AD mouse models used in research, for example, the Tg2576 mouse model, which shows both CAA and plaques as well. There is no novelty in this paper.

Secondly, for introducing an AD mouse model, authors should add the behavior study.

Author Response

Firstly, both mice models have been used in AD studies for years, the characters of both models have already been recorded well and bred with other transgenic mouse models for new characters. Tons of papers have been published. Besides the two models, there are also a lot of AD mouse models used in research, for example, the Tg2576 mouse model, which shows both CAA and plaques as well. There is no novelty in this paper.

We agree with the reviewer that there are many transgenic models well-described and widely used to study the AD pathology, including the APP23 and 5xFAD models. Furthermore, some of those AD transgenic mice also present Aβ deposition in cerebral vessels, although this is a feature much less studied in the context of AD, despite the overlap between AD and CAA pathologies found in the human brain. This was actually the rationale to compare these two models; we wanted to answer which was a proper model to evaluate the vascular contribution in AD (in terms of testing new anti-Aβ drugs including antibodies, evaluating ARIAs, etc) in the future, and also which model should be used to determine the implications of an hemorrhagic phenotype caused by the deposition of cerebrovascular Aβ. We recognize that our study is very methodological and descriptive, but we consider that the characterization of these commonly used experimental models in the AD and CAA field is indispensable to ensure a better translation of pre-clinical results to human clinical trials.

Furthermore, we believe that our study provides interesting and novel results by characterizing the parenchymal and vascular amyloid deposition taking into account the sex and older ages (up to 24 months old), which are features not commonly considered in many reports. In fact, how sex affects different pathological processes has become a crucial topic. Thus, analyzing features associated with AD pathology in different models may serve as a reference to select the appropriate approach depending on the endpoint that needs to be evaluated. For instance, we consider very interesting the fact that a more prominent sex dependent parenchymal Aβ pathology was found in APP23 than in 5xFAD mice, as described in the human pathology. On the other hand, our report provides new findings regarding the vascular Aβ-pathology in the 5xFAD model, which has been a controversial question in the literature. Because the 5xFAD model is a widely used AD model, the profound characterization of its phenotype can always be considered a relevant issue.

Secondly, for introducing an AD mouse model, authors should add the behavior study.

We thank the reviewer for this constructive comment. In this context, we have included some information related to the differences of the transgenic models in terms of behavior and cognition in the Introduction section of the new version of the manuscript:

“Regarding the functional phenotype of these models, behavioral and cognitive studies in the APP23 line described the appearance of first deficits in spatial memory at 3 months of age, which increased progressively with age (Kelly et al., 2003). However, deficits in the other types of memory did not appear until 12 months of age, although a non-linear age-and genotype-dependent behavioral signatures have been found in long-lived APP23 mice (Giménez-Llort et al., 2021). On the contrary, in the 5xFAD model, important cognitive abilities were described in spatial memory tests much earlier in 3-6 months old mice, worsening with age. Also, 5xFAD mice from 9-12 months of age showed a progressive decrease in anxiety and motor deficits (Jawhar et al., 20212).“

Reviewer 3 Report

The study proposed by Mazaruela and collaborators investigates and compares the amyloid deposition in 2 transgenic mouse models of Alzheimer’s disease. In particular, authors focus on the cerebral amyloid angiopathy and deposition of amyloid around brain vessels. Age-related deposition and sex differences are investigated.

This topic is very hot and totally justified. The manuscript is very well written and illustrated, the study is very well designed, results are adequately analyzed and statistical analysis have been performed. To be suitable for the IJMS readership, I would suggest to improve these following points :

The figure 1 is not clear for me. In particular, the presence of WT on the right top corner. In the title, only APP23 and 5xFAD mice are mentioned. Do the authors wanted to write 4 m.o instead of WT ? If this is really WT labelling, please adapt the title and there is no reason that APP23 and 5xFAD are written at the top… And if this is WT labelling, how old is the mouse ? Then, please add female in the title of the figure 1 to make easiest the understanding of the results.

I am wondering if the figure 2 is really necessary. My understanding is that the figure 2 is a merge of the both figures presented in 3A. What are the differences between Figures 2 and 3A?

In the part 2.2, line 2 “figure 5A” should be replaced by “figure 4A”. How old are the mouse used for the figure 4A ? Is it a male or a female ?

The part 2.3 and the figure 5 describe Abeta40 and Abeta42 deposition in APP23 and 5xFAD mice, but if this is male or female should be indicated. The figures numbering should be double checked as the description in the text does not seem to correlate. For example : “Abeta cortical vessels were robustly stained with Abeta40 antibody in APP23 mice (Fig5A2)”. But Fig5A2 shows 4G8 labelling in leptomeningeal vessels.

Anyway, all these results showed in figure 5 are very interesting and I would suggest to summarize them in a table to allow a quick and efficient comparison of Abeta vascular deposition between APP23 and 5xFAD mouse at the levels of leptomeningeal vessels, parenchymal vessels, cortical vessels, capillaries, etc. Authors wrote at the end of this section that “some positive staining was observed in capillaries using the 4G8 and anti-Abeta42 antibodies in APP23 mice…” but this is not clearly indicated in the figure 5. Or maybe “Abeta42” should be replaced by “Abeta40” because the Abeta 42 labelling is very very weak? Do authors have a proof that they are really talking about capillaries ? Did they perform co-incubation with endothelial/capillary markers ? Maybe they are talking about the black arrows in figure 5 ?

Then, always in figure 5, authors observe that Abeta40 is the major form that accumulates around brain microvessels. Do the authors know the ratio Abeta40/42 produced in each transgenic mouse model ?

Why the part 2.4 is only performed in male ? Amyloid deposition seems to be increased in females when compared with males. Authors should justify.

Minor :

Does the low number of animals can be an issue ? Maybe this point needs to be addressed in the conclusion.

In the fifth paragraph of the conclusion, the word “however” appears 3 times in 4 consecutive sentences.

In the “material and methods” section, in the “part 4.6”, line 2, the reference 35 should be formatted.

Author Response

This topic is very hot and totally justified. The manuscript is very well written and illustrated, the study is very well designed, results are adequately analyzed and statistical analysis have been performed.

We thank the reviewer for such a positive and detailed review of our paper, including several excellent suggestions which we have tried to address in this revised version.

To be suitable for the IJMS readership, I would suggest to improve these following points:

The figure 1 is not clear for me. In particular, the presence of WT on the right top corner. In the title, only APP23 and 5xFAD mice are mentioned. Do the authors wanted to write 4 m.o instead of WT? If this is really WT labelling, please adapt the title and there is no reason that APP23 and 5xFAD are written at the top… And if this is WT labelling, how old is the mouse? Then, please add female in the title of the figure 1 to make easiest the understanding of the results.

We apologize for this incomprehensible mistake. The first panels of Fig.1 refer to the ThS staining of brain sections from 16 months old WT mice. Accordingly, we have modified the figure, the title and the text in the figure legend.  

I am wondering if the figure 2 is really necessary. My understanding is that the figure 2 is a merge of the both figures presented in 3A. What are the differences between Figures 2 and 3A?

The reviewer is totally right, the previous figure 2 and figure 3A derived from the same data. However, in the previous Figure 2 (new Fig. 1B) we tried to graphically identify differences between both transgenic lines in terms of number of Aβ deposits (APP23 vs. 5xFAD). On the contrary, in previous Figure 3 (new Fig. 2), the intention was to graphically represent and statistically analyze the potential differences considering age and sex individually for each model (APP23 or 5xFAD separately). In this sense, we have included a sentence in the results section to try to avoid confusions.

We still consider that both figures provide interesting information. In this regard, we suggest saving both figures, although we have merged the previous Fig 2 with the new Fig 1 to keep as much information as possible in a single figure. In fact, we consider that new Fig 1B is only a graphic representation to globally visualize the comparison in the number of Aβ deposits between both models, but the statistical details should be found in new Fig.2. However, if the reviewer or the editors do not approve this merge, we will fully agree to keep Fig 1 and 2 as they originally were or even to remove the second one.

In the part 2.2, line 2 “figure 5A” should be replaced by “figure 4A”. How old are the mouse used for the figure 4A? Is it a male or a female?

We thank the review for noticing this mistake, which has been corrected in the new version of the manuscript. Representative images in Fig 4A (new Fig. 3A) were obtained from brain sections from 16-months old female mice (as in Fig. 1). We have modified the new figure legend 3A to include all the missing information.

The part 2.3 and the figure 5 describe Abeta40 and Abeta42 deposition in APP23 and 5xFAD mice, but if this is male or female should be indicated. The figures numbering should be double checked as the description in the text does not seem to correlate. For example: “Abeta cortical vessels were robustly stained with Abeta40 antibody in APP23 mice (Fig5A2)”. But Fig5A2 shows 4G8 labelling in leptomeningeal vessels.

The sex of the mice selected in previous Fig 5 is indicated in the figure legend (new Fig. 4): “Total Aβ (determined by anti-4G8), Aβ40 and Aβ42 immunodetection analyzed in brain sections of 16-month-old female APP23 (A) and 5xFAD (B) mice”. According to the reviewer's comment, the figure numbering has been revised and modified throughout the text.

Anyway, all these results showed in figure 5 are very interesting and I would suggest to summarize them in a table to allow a quick and efficient comparison of Abeta vascular deposition between APP23 and 5xFAD mouse at the levels of leptomeningeal vessels, parenchymal vessels, cortical vessels, capillaries, etc. Authors wrote at the end of this section that “some positive staining was observed in capillaries using the 4G8 and anti-Abeta42 antibodies in APP23 mice…” but this is not clearly indicated in the figure 5. Or maybe “Abeta42” should be replaced by “Abeta40” because the Abeta 42 labelling is very very weak? Do authors have a proof that they are really talking about capillaries? Did they perform co-incubation with endothelial/capillary markers? Maybe they are talking about the black arrows in figure 5?

The reviewer’s comment made us analyze again the possible Aβ42 positivity in capillaries in new immunohistochemistry analysis. However, we could not confirm our initial perception. Thus, we removed the sentence related to Aβ42 positivity in capillaries from the results sections.

On the other hand, we consider that the 4G8-positivy in capillaries is clear in APP23 brains, in comparison to 5xFAD brains. In fact, Calhoun et al. (1999) already stated that larger vessels were more likely to contain fibrillary amyloid in the APP23 model, although vessels of all sizes, from capillaries to large arteries, were affected to some degree (Ref.# 38).

We have included some red arrowheads in the new Fig.4A1 to show the positive capillaries stained with anti-4G8 antibody (referred in the corresponding figure legend).

Regarding the black arrowheads, we have also noticed that a proper explanation of their meaning was missing in the text. Thus, we have included so in the Figure Legend 5 (new figure legend 4) of the new version of the manuscript. We hope all these changes and the slight modification of the figure (and the figure legend) make them more understandable to the reviewer and the readers.

Then, always in figure 5, authors observe that Abeta40 is the major form that accumulates around brain microvessels. Do the authors know the ratio Abeta40/42 produced in each transgenic mouse model?

We are sorry, but we cannot accurately answer this question using the IHC methodology. However, from our results and previous publications that determined the brain levels of Aβ40 and Aβ42 by ELISA, we can assume that the ratio Aβ40/Aβ42 is much higher in the APP23 model than in the 5xFAD line. 

Why the part 2.4 is only performed in male? Amyloid deposition seems to be increased in females when compared with males. Authors should justify.

Interestingly, in contrast to parenchymal Aβ deposition, no differences in CAA-affected vessels were found between APP23 males and females. In fact, a trend towards higher number of Aβ positive vessels was observed in 24-month-old males compared to age-matched females (new Fig. 3). According to these results, we selected old male mice to evaluate the presence of microbleeds potentially due to the CAA development.

Minor:

Does the low number of animals can be an issue? Maybe this point needs to be addressed in the conclusion.

The low number of animals used in some experiments (eg. MRI analysis) is certainly a limitation of the study. We have included this point in the Discussion section of the revised version of the manuscript. 

In the fifth paragraph of the conclusion, the word “however” appears 3 times in 4 consecutive sentences.

We thank the reviewer for noticing this. We have modified the text to avoid this repetition.

In the “material and methods” section, in the “part 4.6”, line 2, the reference 35 should be formatted.

We have included the correct number of the reference, as well as the proper cite in the Reference List (#76).

Reviewer 4 Report

COMMENTS FOR THE AUTHORS

Animal models are of great interest in studying mechanisms and potential treatments for Cerebral Amyloid Angiopathy (CAA). To be “translational” and to impact on clinical practice, an animal model should reproduce at least one of the pathological processes seen in human CAA. A full translational model could serve to identify novel molecular, cellular and physiological mechanisms of the pathological process. Although CAA was first described almost a century ago, progress in deciphering its underlying pathological mechanisms has been hindered by the lack of reliable animal models. Since rodents generally do not develop CAA spontaneously, not even at very old ages, a variety of transgenic mouse models have been introduced that, similar to CAA in humans, develop either Aβ-CAA only or both Aβ-CAA and parenchymal amyloid, or primarily parenchymal amyloid. Transgenic mouse models displaying a variable degree of CAA, short life span, low expense and easy genetic manipulations have emerged, thus providing us with some key findings in the disease pathophysiology. However, since no single animal model for CAA completely resembles human CAA, the integration between experimental results and clinical data from large observational studies are mandatory to understand the mysteries of CAA and to develop effective therapeutic strategies. The present manuscript appears to be an extensive, well-written and well-conceived study addressing the translational use of transgenic murin models to investigate the overlap between vascular and parenchimal Aβ deposition in the brain. Specifically, the Authors have highlighted the impact of age and sex on disease development. The selected topic is of particular interest because it will contribute to the translation of pre-clinical in vivo evidence to clinical practice, in the field of CAA and also AD (Alzheimer’s disease) treatment.

Only minor changes and typo corrections are requested.

Minor changes:

  • Figure 2: Please, insert “Ab" in Y axes label;
  • Figure 3: the present Figure is not self-explaining, because some details -regarding statistical analyses of the reported results- can be deduced only by reading Paragraph 4.9. See, for example, the difference between #p< 0.01 and **p< 0.01 and the reference to the effects of age (A) and sex (S);
  • Legend to Figure 5: the references to (1), (2) and (3) (indicating 4G8, Aβ40 and Aβ42 immuno-detections) are superfluous. Please, pick-out better the scale bars indicating 200 μm, 100 μm and 40 μm.

Typo errors:

  • Page 1, Abstract: please, change from “…to evaluate progression…with age and sex…” to “…to evaluate progression…in relation to age and sex...”;
  • Page 1, Abstract: please, change from “…disease-modifying therapies before its…” to “…disease-modifying therapy before its…”;
  • Page 2, last sentence of Introduction: “The study and comparison of the two models…”
  • Page 5, paragraph 2.2: “As shown in Figure 4A…”
  • Page 9, line 1: “…which also make difficult…”
  • Page 10: “…has been scarce and controversial

Author Response

We totally agree with the reviewer’s considerations about the need of further translational investigation in the CAA field. We thank the reviewer for his/her nice comments about our study.

Minor changes:

Figure 2: Please, insert “Ab" in Y axes label;

We have included the term “ThS (+)” in the Y axes (new Fig.1B) to maintain the uniformity with Fig.2. In the new figure legend 1B, we have included the corresponding explanation: “Data are expressed as the number of ThS-positive deposits (fibrillar Aβ deposits) per area”.

Figure 3: the present Figure is not self-explaining, because some details -regarding statistical analyses of the reported results- can be deduced only by reading Paragraph 4.9. See, for example, the difference between #p< 0.01 and **p< 0.01 and the reference to the effects of age (A) and sex (S)

The reviewer is totally right and we apologize for the mistakes in the figure legend 3 (new Fig. 2). We have tried to address this point by adding the correct information in new figure legend 2 and 3. We consider that these modifications make both figures self-explaining and understandable to the reader in terms of evaluating the age and sex effects and the differences among the groups compared in each case.  

Legend to Figure 5: the references to (1), (2) and (3) (indicating 4G8, Aβ40 and Aβ42 immuno-detections) are superfluous. Please, pick-out better the scale bars indicating 200 μm, 100 μm and 40 μ

We have slightly modified the previous Fig.5 (new Fig. 4), including the scale bars, to make it more understandable. We have also extended the corresponding figure legend to include relevant information regarding to panels, arrowheads and scale bars.  

Typo errors:

Page 1, Abstract: please, change from “…to evaluate progression…with age and sex…” to “…to evaluate progression…in relation to age and sex...”;

Page 1, Abstract: please, change from “…disease-modifying therapies before its…” to “…disease-modifying therapy before its…”;

Page 2, last sentence of Introduction: “The study and comparison of the two models…”

Page 5, paragraph 2.2: “As shown in Figure 4A…”

Page 9, line 1: “…which also make difficult…”

Page 10: “…has been scarce and controversial

We thank the reviewer for noticing all these mistakes, which we have addressed in the new version of the manuscript.

Round 2

Reviewer 1 Report

Thank you for your response, accepted!!

Author Response

Thank you